# Evaluation of the Effect of an α-Adrenergic Blocker, a PPAR-γ Receptor Agonist, and a Glycemic Regulator on Chronic Kidney Disease in Diabetic Rats

**DOI:** 10.3390/ijms252111372

**Published:** 2024-10-23

**Authors:** Jorge Morones, Mariana Pérez, Martín Muñoz, Esperanza Sánchez, Manuel Ávila, Jorge Topete, Javier Ventura, Sandra Martínez

**Affiliations:** 1Department of Morphology, Basic Sciences Center, Universidad Autónoma de Aguascalientes, Aguascalientes 20100, Mexico; al239857@edu.uaa.mx (J.M.); al77006@edu.uaa.mx (M.P.); esperanza.sancheza@imss.gob.mx (E.S.); manuel.avila@edu.uaa.mx (M.Á.); 2Department of Chemistry, Basic Sciences Center, Universidad Autónoma de Aguascalientes, Aguascalientes 20100, Mexico; humberto.munozo@edu.uaa.mx; 3Family Medicine Unit 8, Instituto Mexicano del Seguro Social (IMSS), Aguascalientes 20180, Mexico; 4Department of Nephrology, Regional General Hospital No. 46, Instituto Mexicano del Seguro Social (IMSS), Guadalajara 44910, Mexico; Fernando.topete@live.com.mx; 5Department of Microbiology, Basic Sciences Center, Universidad Autónoma de Aguascalientes, Aguascalientes 20100, Mexico

**Keywords:** chronic kidney disease, diabetic nephropathy, hyperglycaemia, tamsulosin, pioglitazone, linagliptin

## Abstract

Diabetic nephropathy (DN) is a globally widespread complication of *diabetes mellitus* (DM). Research indicates that pioglitazone and linagliptin mitigate the risk of DN by reducing inflammation, oxidative stress, and fibrosis. The role of tamsulosin in DN is less studied, but it may contribute to reducing oxidative stress and inflammatory responses. The protective effects of combining pioglitazone, linagliptin, and tamsulosin on the kidneys have scarcely been investigated. This study examines the individual and combined effects of these drugs on DN in Wistar rats. Diabetic rats were treated with tamsulosin, pioglitazone, and linagliptin for six weeks. We assessed food and water intake, estimated glomerular filtration rate (eGFR), histological markers, urea, creatinine, glucose, NF-κB, IL-1, IL-10, TGF-β, and Col-IV using immunofluorescence and qPCR. The DN group exhibited hyperglycaemia, reduced eGFR, and tissue damage. Tamsulosin and linagliptin improved eGFR, decreased urinary glucose, and repaired tissue damage. Pioglitazone and its combinations restored serum and urinary markers and reduced tissue damage. Linagliptin lowered serum creatinine and tissue injury. In conclusion, tamsulosin, linagliptin, and pioglitazone demonstrated renoprotective effects in DN.

## 1. Introduction

*Diabetes mellitus* (DM) encompasses metabolic disorders characterized by hyperglycaemia and is generally classified based on insulin production [1]; over the long term, elevated blood glucose levels can lead to retinopathy and neuropathy [2,3], as well as being one of the main triggers in the development of chronic kidney disease (CKD) [4,5].

In 2021, DM affected 537 million people worldwide. According to the International Diabetes Federation (IDF), this number is projected to rise to 643 million by 2030 and 783 million by 2045 if comprehensive prevention and medical control strategies are not implemented [6]. CKD saw a 10% annual increase in 2022, affecting approximately 800 million people globally [7]. A significant cause of CKD is DN [8,9]. DN is a high-mortality disease caused by the presence of DM, whether type I or type II, and is characterized by alterations in normal renal function and morphology [10]. Additionally, DN is the leading cause of end-stage renal disease (ESRD) [11,12], affecting one in five people with type 2 DM and one in three with type 1 DM worldwide [13]. Approximately 30–40% of people with DM develop DN [14], and the rising prevalence of DM suggests corresponding increases in DN. Patients reaching the ESRD stage have a mortality rate of 53% [15]. DN is characterized by the convergence of various pathways for disease progression, such as TGF-β, NF-κB, and PKC. DN is a complex disease that is not yet fully understood [16], resulting in therapies that have only slowed its progression. The clinical symptoms of DN include hyperglycaemia, the accumulation of serum metabolites such as BUN and creatinine, microalbuminuria in most cases, and a decrease in eGFR [17,18]. The main morphological alterations include mesangial proliferation and expansion, glomerular hypertrophy, tubulointerstitial fibrosis, glomerulosclerosis, and thickening of the glomerular basement membrane [12,19,20,21,22,23].

Oxidative stress plays a crucial role in the progression of DN because hyperglycaemia induces the formation of advanced glycation end products (AGEs) [24], leading to the subsequent release of mediators such as angiotensin II and TGF-β, which have been reported to have a significant impact on DN [25,26]. Additionally, hyperglycaemia induces the production of reactive oxygen species (ROS). Inflammation and dysregulation between the synthesis and degradation of the extracellular matrix (ECM) are also factors that contribute to the development of the disease, hyperfiltration, loss of renal parenchyma, and ultimately the development of ESRD [27,28].

Pioglitazone, a drug belonging to the thiazolidinedione group [29], acts by stimulating nuclear PPAR-γ receptors, whose function has been better characterized in recent years [30]. Several effects of the agonism of these receptors have been described, such as the regulation of glucose metabolism [31], reduction in insulin resistance [32], and the inhibition of inflammatory signaling pathways via the transcription factor NF-κB [33]. Additionally, studies have shown that PPAR-γ agonists exhibit antiproliferative effects, enhance glucose sensitivity in inflammation, and have antifibrotic effects in both the lungs and kidneys [34], the latter due to the presence of PPAR-γ in various renal parenchymal cells, such as tubular, interstitial, endothelial, mesangial, and juxtaglomerular cells [33].

Linagliptin is a xanthine derivative [35], belongs to dipeptidyl peptidase-4 (DPP-4) inhibitors, and is an enzyme that degrades incretin hormones such as glucagon-like peptide-1 (GLP-1) and glucose-dependent insulinotropic polypeptide (GIP), which stimulate insulin release from pancreatic β-cells and regulate glucose metabolism, among other functions [36,37]. Linagliptin is primarily used in the treatment of type 2 DM [38,39], although studies also indicate its use in atherosclerosis, dyslipidemias, and hypertension [40]. Currently, its potential use in neurological and immune system diseases is under investigation [41,42]. The use of this type of drug has been reported to have renoprotective effects independent of glycemic levels [43,44,45], likely because DPP-4 has many substrates that are still unknown [46].

Tamsulosin is a methoxybenzenesulfonamide [47], classified as a selective adrenergic antagonist of α1A and α1D receptors [48,49]. It was the first drug used in the USA in 1997 for the treatment of benign prostatic hyperplasia (BPH), characterized by urinary tract obstruction [50]. Additionally, it has been used in the treatment of abdominal aortic aneurysm and ureteral lithiasis [51,52]. However, its use in CKD and DN remains unknown.

The kidneys are highly innervated by the sympathetic nervous system (SNS) [53,54], which significantly modulates renal functions such as renal blood flow and, consequently, the eGFR [55,56]. Additionally, the SNS influences the reabsorption of sodium and water at the renal tubular level. Multiple α-adrenergic receptors have been identified in the renal vasculature and tubular cells [57]. Studies have shown that exacerbated SNS activity in CKD is associated with tubular damage, renal fibrosis, and inflammation [58,59]. This inflammation plays a crucial role in the progression of CKD [60]. Moreover, immune cells like macrophages, through α-adrenergic receptors, are activated by norepinephrine and can stimulate the release of TNF-α [61], which has been linked to the progression of DN [62]. It is suggested that tamsulosin could have significant effects in preventing the development of DN by mitigating renal dysfunctions triggered by SNS overactivation.

The aim of this study was to analyze the effect of tamsulosin, pioglitazone, and linagliptin on DN, using individual and combined doses.

## 2. Results

### 2.1. Evaluation of Renal Index, Water and Food Intake, and eGFR

The renal index (KI) is a macroscopic indicator of renal damage, calculated by comparing the total body weight of the animal to the weight of both kidneys. Its purpose is to identify changes in renal mass. The KI results were significantly higher in the DN group (10.73 mg/g; *p* < 0.001) compared to the I group (6.08 mg/g), showing an increase in kidney weight relative to body weight. Conversely, all other groups—TS (7.93 mg/g), PIO (6.27 mg/g), LG (8.08 mg/g), LG + PIO (6.46 mg/g), and LG + TS + PIO (7.83 mg/g), except for the LG + TS group (8.48 mg/g)—showed a significant decrease in KI (*p* < 0.05, *p* < 0.001) compared to the DN group. The PIO and LG + PIO groups exhibited KI values very close to those of the intact group (Figure 1a). Regarding water intake, the DN group showed a higher consumption (499.29 mL/day; *p* < 0.001) compared to the healthy rats (I) group (158.57 mL/day). The TS (447.14 mL/day), LG (471.43 mL/day), and LG + TS (481.43 mL/day) groups did not show significant changes in water consumption compared to the DN group (499.29 mL/day). In contrast, the PIO (214 mL/day; *p* < 0.001), LG + PIO (283.57 mL/day; *p* < 0.001), and LG + TS + PIO (347.14 mL/day; *p* < 0.001) groups significantly reduced water consumption compared to the DN group, with the PIO group showing the lowest intake (Figure 1b). Similarly, we evaluated food intake across the different treatment groups. The TS (100.57 g/day), PIO (99 g/day), LG (94.57 g/day), LG + PIO (94.29 g/day), and LG + TS + PIO (115.86 g/day) groups significantly reduced food consumption compared to the DN group (139.14 g/day; *p* < 0.001), with no significant effect observed in the LG + TS group (131.29 g/day) (Figure 1c). Finally, when evaluating eGFR, we observed that the groups receiving tamsulosin TS (0.96 mL/min; *p* < 0.05), pioglitazone PIO (1.17 mL/min; *p* < 0.01), or linagliptin LG (1.01 mL/min; *p* < 0.01) showed improved eGFR compared to the DN group (0.35 mL/min). Notably, among the three drugs used, pioglitazone had slightly higher eGFR values, although not reaching the levels seen in the healthy rats (I, 3.0 mL/min). Regarding the combination of drugs, only the LG + TS + PIO group showed a higher eGFR (1.21 mL/min) compared to the LG + TS (0.85 mL/min) and LG + PIO (1.19 mL/min) groups (Figure 1d).

### 2.2. Effect of Tamsulosin, Pioglitazone, and Linagliptin on Renal Damage Markers

To determine whether these drugs could aid in the recovery of renal function, various serum markers of renal damage were evaluated. Fasting blood glucose (FBG), creatinine, blood urea nitrogen (BUN), and urea levels were measured. The DN group exhibited an FBG concentration of 697.4 mg/dL compared to the I group (106.6 mg/dL; *p* < 0.0001). The TS (416.2 mg/dL), LG (518.4 mg/dL), and PIO (344.83 mg/dL) groups showed reduced FBG levels, with the PIO group showing significant differences compared to the DN group (*p* < 0.01). Combination treatments such as LG + PIO (408.8 mg/dL) and LG + TS + PIO (477.33 mg/dL) also reduced FBG levels, though not significantly compared to the DN group. The LG + TS group (699.8 mg/dL) showed no change in FBG levels (Figure 2a). Regarding creatinine, BUN, and serum urea, the DN group had concentrations of 0.87 mg/dL, 35.6 mg/dL, and 78.97 mg/dL, respectively, which were higher than those in the healthy group (I) (creatinine: 0.23 mg/dL, BUN: 20.4 mg/dL, urea: 53.93 mg/dL). All treated groups showed reduced creatinine levels, with significant reductions observed in the PIO (0.64 mg/dL) and LG (0.62 mg/dL) groups compared to the DN group (*p* < 0.05) (Figure 2b). In contrast to creatinine, only the PIO group showed significantly lower BUN (23.67 mg/dL), and urea (50.65 mg/dL) levels compared to the DN group (*p* < 0.05, *p* < 0.01). The TS, LG, LG + PIO, LG + TS, LG + PIO, and LG + TS + PIO groups did not show significant differences.

### 2.3. Effect of Tamsulosin, Pioglitazone, and Linagliptin on Urine Glucose, BUN, and Urea

The concentrations of glucose, creatinine, BUN, and urea in the urine of different treated rat groups are shown in Figure 3. The DN group had a glucose concentration of 1995.37 mg/dL, which was higher than the healthy control (I). In the TS (1433.33 mg/dL; *p* < 0.0001), PIO (733.33 mg/dL; *p* < 0.0001), and LG (1633.33 mg/dL) groups, the PIO group reduced glucose concentration by up to 50% compared to the DN group. Among the combination treatments, only the LG + PIO group (1225 mg/dL; *p* < 0.0001) showed a significant reduction.

Creatinine levels improved in the TS (58.17 mg/dL; *p* < 0.01), PIO (69.08 mg/dL; *p* < 0.0001), LG (59.89 mg/dL; *p* < 0.001), LG + PIO (58.84 mg/dL; *p* < 0.001), and LG + TS + PIO (51.32 mg/dL; *p* < 0.05) groups. The LG + TS group (11.33 mg/dL) had a lower concentration than even the DN group (35.42 mg/dL).

Regarding BUN and urea, only the PIO (363 mg/dL; *p* < 0.01 and 776.82 mg/dL; *p* < 0.05, respectively) and LG + PIO (395.75 mg/dL and 846.91 mg/dL; *p* < 0.05, respectively) groups significantly reduced their concentrations compared to the DN group.

### 2.4. Effect of Tamsulosin, Pioglitazone, and Linagliptin on Renal Histopathology

Hematoxylin–eosin (H&E) staining was performed to evaluate tissue damage, and Masson’s trichrome and Sirius red staining were used to visualize the collagen fibers. The healthy control group (I) showed no abnormal changes in renal histopathology. The DN group induced with STZ exhibited reduced urinary space (12.57%; *p* < 0.0001), increased mesangial expansion, glomerular hypertrophy, focal and segmental glomerulosclerosis characterized by collagen I deposition (36.73%; *p* < 0.0001), GBM thickening (29.53%; *p* < 0.0001), tubulointerstitial fibrosis (COL I, 8.01% and COL III, 5.36%; *p* < 0.0001), and glomerular alteration. The TS group showed less glomerular hypertrophy, and a larger urinary space (15.94%) compared to the DN group, along with reduced collagen I in the glomerulus (30.50%; *p* < 0.001) and GBM (22.84%; *p* < 0.001), as well as collagen I in the tubulointerstitial area (COL-I, 5.93% and COL-III, 4.39%; *p* < 0.01). In the PIO and LG groups, there was a recovery of normal glomerular and tubulointerstitial structure, with an increase in urinary space (24.46%; *p* < 0.0001 and 18.66%; *p* < 0.05, respectively), reduced collagen I in the glomerular area (18.99% and 27.95%; *p* < 0.0001, respectively) and GBM (14.48% and 20.26%; *p* < 0.0001, respectively), as well as decreased collagen I and III in the tubulointerstitial area (PIO: COL I, 1.65%; *p* < 0.0001, COL III, 2.07%; *p* < 0.0001 and LG: COL I, 5.89%; *p* < 0.01, COL III, 2.72%; *p* < 0.0001). Regarding the combination of drugs, the LG + TS, LG + PIO, and LG + TS + PIO groups showed results similar to the PIO and LG groups: urinary space (15.27%, 22.75%, *p* < 0.0001, 21.87%, *p* < 0.0001, respectively), collagen I in the glomerular area (25.72%, *p* < 0.0001, 19.37%, *p* < 0.0001, 19.89%, *p* < 0.0001, respectively), GBM (20.69%, *p* < 0.0001, 15.74%, 21.22%, *p* < 0.0001, respectively), and collagen I and III (COL I, 5.14%, *p* < 0.0001, COL I, 2.84%, *p* < 0.0001, COL I, 2.04%, *p* < 0.0001, respectively, and COL III, 2.86%, *p* < 0.0001, COL III, 1.61%, *p* < 0.0001, COL III, 2.51%, *p* < 0.0001, respectively).

### 2.5. Anti-Inflammatory Effect of Tamsulosin, Pioglitazone, and Linagliptin on DN

The anti-inflammatory effect of the different drugs used was evaluated by immunofluorescence and RT-qPCR. Immunofluorescence analysis showed a significant increase in NF-κB- and IL-1β-positive cells in the DN group (844.44 cells/mm^2^, *p* < 0.001, 735.24 cells/mm^2^, *p* < 0.0001) compared to the I group (352.68, 373.60 cells/mm^2^). All groups showed a reduction in the number of NF-κB- and IL-1β-positive cells. However, we saw that the PIO (387.05 cells/mm^2^, *p* < 0.001), LG + PIO (394.52 cells/mm^2^, *p* < 0.001), and LG + PIO + TS (404.98 cells/mm^2^, *p* < 0.01) groups had NF-κB-positive cells similar to the intact group (Figure 4). Regarding IL-1β, the PIO (363.14 cells/mm^2^, *p* < 0.0001), LG (397.51 cells/mm^2^, *p* < 0.0001), LG + PIO (479.7 cells/mm^2^, *p* < 0.0001), and LG + PIO + TS (494.64 cells/mm^2^, *p* < 0.0001) groups were similar to the intact group. On the other hand, when evaluating the presence of IL-10-positive cells, we observed that the DN group (665.99 cells/mm^2^, *p* < 0.0001) had lower values compared to the I group (1376.29 cells/mm^2^). The group treated with pioglitazone (1133.13 cells/mm^2^, *p* < 0.0001) showed a higher number of IL-10-positive cells than the groups treated with tamsulosin (855.15 cells/mm^2^) and linagliptin (897.16 cells/mm^2^), whether used alone or in combination. However, compared to the DN group, the LG + PIO (1024.13 cells/mm^2^, *p* < 0.001) and LG + PIO + TS (1022.94 cells/mm^2^, *p* < 0.001) groups also showed significant increases (Figure 5). To correlate these immunofluorescence observations, we analyzed the relative gene expression levels of these markers. The DN group exhibited a significant increase in NF-κB (1.67-fold, *p* < 0.05) and IL-1β (3.03-fold, *p* < 0.0001) but not in IL-10 (0.46-fold, *p* < 0.0001) compared to the control group (I) (1.04-fold, 1.03-fold, and 1.61-fold, respectively). Regarding the drugs, all groups except the LG + TS group showed lower expression levels of NF-κB and IL-1β (2.01-fold, *p* < 0.001, 2.13-fold, *p* < 0.001). Concerning IL-10, the drugs promoted the expression of this cytokine, with the PIO group (1.42-fold, *p* < 0.0001) showing a higher expression level compared to the DN group (0.46-fold) (Figure 6 and Figure 7).

### 2.6. Effect of Tamsulosin, Pioglitazone, and Linagliptin on Fibrogenic Markers in DN

To determine whether the drugs used have any effect on fibrosis, the number of TGF-β-positive cells and the area of collagen IV positivity were analyzed by immunofluorescence, and relative gene expression was measured (Figure 8 and Figure 9). The results showed that the DN group had a significant increase in TGF-β-positive cells (1255.29 cells/mm^2^, *p* < 0.0001) and collagen IV-positive area (15.47% positive area, *p* < 0.0001) compared to the intact group (342.22 cells/mm^2^, 3.85% of area). On the other hand, we observed that the groups of rats that received the drugs individually significantly reduced the number of TGF-β-positive cells and the collagen IV-positive area. However, the group that showed the greatest reduction in TGF-β-positive cells and collagen IV-positive area was the group that received pioglitazone (337.73 cells/mm^2^, 6.30% of area, *p* < 0.0001). The groups of rats that received the combination of drugs (LG + TS (958.12 cells/mm^2^, *p* < 0.05, 11.54% of area, *p* < 0.05), LG + PIO (396.01 cells/mm^2^, *p* < 0.0001, 6.31% of area, *p* < 0.0001), and LG + TS + PIO (657.53 cells/mm^2^, *p* < 0.0001, 8.40% of area, *p* < 0.0001)) showed a reduction in TGF-β-positive cells and collagen IV-positive area compared to the DN group. Notably, among the groups listed above, LG + PIO and LG + TS + PIO presented values close to the I group (342.22 cells/mm^2^, 3.85% of area).

On the other hand, the analysis of the gene expression of TGF-β and collagen IV in the DN group (1.74-fold, *p* < 0.01, 3.26-fold, *p* < 0.0001, respectively) showed a significant increase compared to the I group (1.0-fold, 1.16-fold, respectively). The drugs administered, either alone or in combination, showed a decrease in TGF-β gene expression, except for the LG + TS group (1.35-fold). Regarding collagen IV, a similar effect was observed (2.24-fold, *p* < 0.0001). However, the PIO (1.04-fold, *p* < 0.05, 1.0-fold, *p* < 0.0001, respectively) and LG + PIO (1.12-fold, *p* < 0.05, 1.30-fold, *p* < 0.0001, respectively) groups showed expression levels like the intact group (1.0-fold, 1.16-fold, respectively).

### 2.7. Effect of Tamsulosin, Pioglitazone, and Linagliptin on NRF2 and HO-1 in DN

Finally, we examined the expression of crucial genes in signaling pathways such as NRF2 and HO-1 to elucidate the underlying mechanism by which these drugs mitigate oxidative stress and inflammation in DN (Figure 10). The results showed that tamsulosin (1.02-fold, *p* < 0.01, 1.0-fold, *p* < 0.05), pioglitazone (1.67-fold, *p* < 0.0001, 1.22-fold, *p* < 0.001, respectively), and linagliptin (1.0-fold, *p* < 0.05, 1.02-fold, *p* < 0.05) significantly increased the relative gene expression of NRF2 and HO-1 compared to the DN group (0.51-fold, *p* < 0.01, 0.61-fold, *p* < 0.05). Pioglitazone administration (PIO, 1.67-fold, *p* < 0.0001, 1.22-fold, *p* < 0.001) increased NRF2 and HO-1 expression compared to the other drugs administered. Additionally, among the groups that received the combination of drugs, the LG + PIO group (1.96-fold, *p* < 0.0001) showed a higher relative gene expression of NRF2 compared to the DN and intact groups (1.06-fold). Regarding HO-1, the LG + PIO (1.19-fold, *p* < 0.01) and LG + TS + PIO (1.126-fold, *p* < 0.001) groups showed higher relative gene expression of HO-1 compared to the DN group (0.61-fold, *p* < 0.05).

## 3. Discussion

DN is a globally distributed disease [14], with expected increases in incidence associated with future rises in DM [15]. Additionally, since DN is the leading cause of ESRD [11,12], it is essential to seek new strategies to improve the quality of life of affected individuals. Although some studies mention the use of pioglitazone in the treatment of DN, comparing it with an α-adrenergic blocker (for which there is practically no information on its therapeutic effect in DN) and a DPP4 inhibitor remains a partially unexplored field. The aim of this study was to investigate the antifibrotic, anti-inflammatory, and antioxidant effects of the drugs pioglitazone, tamsulosin, and linagliptin in the treatment of DN in a STZ-induced Wistar rat model.

DN was induced with STZ [63], whose mechanism of action is based on nitric oxide donation to pancreatic islets, DNA alkylation and cross-linking, stimulation of poly-ADP ribosylation, and the uncoupling of glucose access to pancreatic β-cells via GLUT 2 [64]. Three days later, these animals exhibited FBG levels greater than 13.9 mmol/L [65], as well as increased water and food intake and KI compared to the I group. Additionally, the DN model was corroborated by histological findings, including glomerular hypertrophy, mesangial proliferation and expansion, focal and segmental glomerulosclerosis, GBM thickening, and tubulointerstitial fibrosis.

A study in STZ-induced diabetic rats demonstrated that 8-week treatment with pioglitazone was able to reduce the polyphagia and polydipsia associated with DM, correlating with a recovery of adipocyte mass compared to the diseased group, and significantly improving classic parameters such as hyperglycemia and hypoinsulinemia [66]. Another study on diabetes in mice induced with a high-sucrose and high-fat diet showed that 14-week treatment with anagliptin, an analog of linagliptin, in combination with the hormone leptin, resulted in a reduction in diabetes-associated polyphagia, as well as protection via stimulation of the transcription factor STAT3 [67]. DN is characterized by increased food (polyphagia) and water (polydipsia) intake due to the inability to utilize blood glucose. In this context, these two parameters were significantly higher in the DN group rats in our study. Regarding water consumption, this was significantly lower in the group treated with pioglitazone compared to the DN group; however, the TS and LG groups showed no differences. Concerning the combined groups, the LG + PIO and LG + TS + PIO groups also achieved significant reductions compared to the DN group, while the LG + TS group showed no effect. Additionally, food consumption was significantly lower in all groups except the LG + TS group.

DN is also characterized by an increase in KI, a factor that proves the relationship between kidney weight and animal weight [11]. This is due to body weight loss and an increase in renal mass in DN, a pattern observable in fibrotic kidneys [65]. A study conducted in STZ-induced DN rats treated with pioglitazone for 8 weeks showed that the decrease in KI induced a notable improvement in DN, associating it with the increased expression of podocalyxin, a protein related to podocyte architecture, and protection against disease progression [68]. Other studies in STZ-induced diabetic rats treated with sitagliptin, an analog of linagliptin [69], showed that decreases in KI compared to the diabetic group were associated with functional and histopathological improvements through inhibition of the TGF-β/Smad signaling pathway and stimulation of the PI3K/AKT pathway [70,71]. These findings are like our experiments, as the DN group showed a significant increase compared to the I group. Additionally, rats treated with pioglitazone, tamsulosin, and linagliptin, as well as the combinations LG + PIO and LG + TS + PIO, showed significant decreases compared to the DN group.

eGFR is a factor that allows the evaluation of the volume filtered by the kidneys per unit of time and is a universal standard measure used in the assessment of renal health [72]. Additionally, according to *Kidney Disease Improving Global Outcomes* (KDIGO), a decrease of more than 50% from normal values is an important characteristic of CKD and, of course, DN [5,12,73]. There is evidence that the restoration of eGFR in animal models of DN is a vital factor in the functional recovery of renal health [74,75]. This is like our eGFR results, where all applied drugs caused a significant partial recovery compared to the DN group. Although the values did not reach those of the I group, a clear upward trend and, therefore, functional recovery can be seen.

Currently, there is evidence supporting the relationship between excessive sympathetic system activation and its association with CKD and DN, due to the modulation of eGFR, renal blood flow, activation of the renin–angiotensin system (RAS), and sodium and water reabsorption [53,55,57]. Additionally, the presence of α1-adrenergic receptors at the afferent and efferent arterioles of the renal corpuscle’s vascular pole is implicated in the regulation of eGFR [57]. Thus, tamsulosin could potentially act at this level by inhibiting smooth muscle contraction and regulating normal eGFR levels. This could elucidate the possible mechanism of action of this drug and, therefore, explain the relationship between the eGFR results and the administration of tamsulosin in DN.

In pre-hypertensive rats administered with pioglitazone, a reduction in blood pressure was achieved through nitric oxide stimulation, inhibition of AT2 receptors (angiotensin type 2 receptor), and modulation via the NRF2 pathway [76]. This is consistent with our results, where groups treated with pioglitazone alone or in combination showed significant increases in eGFR compared to the DN group.

Similarly, studies in type 1 diabetic murine models have shown that linagliptin exerts vasodilatory effects through interaction with endothelial nitric oxide synthase (eNOS) and caveolin-1 (CAV-1), a protein that negatively regulates eNOS activity [77]. Additionally, evidence of increased eGFR produced by GLP-1, which is promoted by linagliptin use [78], aligns with our observed results in groups administered with linagliptin. However, the LG + TS combination did not produce any effect.

Hyperglycemia in DN is a trigger for damage through the induction of ROS at the mitochondrial level [13]. The importance of glycemic control in DN is such that various treatments focus on this parameter as the main therapeutic target [79,80]. In a study conducted on diabetic rats, the administration of pioglitazone for 6 months resulted in significant reductions in serum glucose compared to the diabetic group [81]. Another study on Otsuka Long-Evans Tokushima fatty (OLETF) diabetic rats also proved that hyperglycemia was significantly reduced by pioglitazone administration for 2 weeks, which was associated with attenuation of glomerular hyperfiltration and restoration of normal macula dense function [82].

Regarding our results, the groups treated with pioglitazone significantly reduced FBG levels compared to the DN group, although they did not reach normal levels (less than 135 mg/dL) [83]. This is due to its mechanism of action: when pioglitazone binds to its nuclear receptor PPARγ, it forms a heterodimer with the retinoic receptor (RXR). This complex is translocated to promoter regions of DNA known as peroxisome proliferator response elements (PPREs) to induce the transcription of genes primarily related to insulin sensitization; lipid and carbohydrate metabolism, such as lipoprotein lipase, fatty acid transport proteins, and LDL oxidase; and enzymes involved in fatty acid reutilization like glycerol kinase and phosphoenolpyruvate carboxykinase [84]. This helps optimal communication between the liver, muscle, and adipose tissue, which are the most insulin-sensitive organs, leading to better glucose utilization [84]. On the other hand, linagliptin also produced a reduction, although not significant, in FBG levels compared to the DN group. A study conducted on db/db mice treated for 16 weeks with empagliflozin (a sodium–glucose transporter inhibitor) in combination with linagliptin showed a significant decrease in FBG levels compared to the vehicle, which is like the results obtained in our model [85]. Additionally, tamsulosin also produced a reduction, although not significant, in FBG levels. Relating to the combination of drugs, the LG + PIO and LG + TS + PIO groups also produced reductions in glycaemia levels, although not significant. However, the LG + TS group showed no effect compared to the DN group.

Disruptions in the glomerular filtration barrier in DN have been associated with the accumulation of serum markers such as creatinine and urea [11,86]. A study involving STZ-induced diabetes proved that the administration of pioglitazone and vildagliptin, the latter being an analog of linagliptin, resulted in greater reductions in markers like serum creatinine and urea compared to individual administration in diseased groups [87]. These serum parameters are influenced by eGFR, with their concentrations being similar in plasma and urine [88]. In our results, creatinine levels were significantly reduced by the PIO, LG, LG + TS, and LG + TS + PIO groups compared to the DN group. However, the TS and LG + PIO groups also showed reductions, though not significant. Regarding BUN and serum urea, only the group administered with pioglitazone alone showed a significant decrease compared to the DN group. Nonetheless, the TS, LG + PIO, and LG + TS + PIO groups also exhibited a trend towards reduction, although not significant.

Urinary glucose has been documented as a marker of tubular dysfunction, and its analysis can be used to determine tubular damage in the progression of renal injury [89]. It is also a common parameter in patients with DM. In this study, the DN group showed high levels of urinary glucose compared to the I group. The drugs used were related to the FBG levels obtained, where those that reduced FBG levels also reduced urinary glucose levels (significant reductions in the TS, PIO, LG + TS, LG + PIO, and LG + TS + PIO groups), suggesting the tubular renoprotective role of these drugs.

Creatinine is a normal waste product of the body, produced by the metabolism of creatine phosphate [90], freely filtered through the glomerulus [91], and its excretion is directly related to eGFR. Thus, the administration of tamsulosin (*p* < 0.01), pioglitazone (*p* < 0.0001), linagliptin (*p* < 0.001), and the combinations LG + PIO (*p* < 0.001) and LG + TS + PIO (*p* < 0.05) produced significant increases compared to the DN group, except for the LG + TS group. This could be due to the logarithmic relationship between eGFR and creatinine concentration [92], so even though linagliptin partially restored eGFR, this was not revealed quantitatively in urine creatinine concentrations.

Regarding BUN and urea, these are not freely filtered through the glomerulus as they are reabsorbed in the proximal convoluted tubule, with approximately 30–50% of the plasma urea load being excreted [88]. Consequently, the DN group showed a significant increase compared to the control group (I), suggesting a probable tubular dysfunction. On the other hand, drug administration managed to reduce this parameter, although only the groups treated with pioglitazone (PIO) and the combination of linagliptin and pioglitazone (LG + PIO) produced significant reductions compared to the DN group, even showing no significant differences with the control group, suggesting a possible protective role at the tubular level.

Renal damage in DN is characterized by the overlap of numerous signaling pathways, such as those triggered by TGF-β, NF-κB, cellular mediators, hemodynamic, hormonal, and metabolic changes [12]. These pathways lead to uncontrolled repair over time, resulting in elevated production of ECM proteins. This process generally begins with renal injury due to hyperglycemia and later production of reactive oxygen species (ROS), which activate the immune system and lead to leukocyte infiltration in the kidney, as well as the secretion of inflammatory cytokines such as IL-1β and IL-17 [60]. Subsequently, activated and differentiated interstitial myofibroblasts (originating from fibroblasts and pericytes) initiate and perpetuate excessive ECM production [93]. Tubular epithelial cells initially suffer apoptosis due to ROS-induced damage; however, when this issue persists over time, these cells undergo epithelial–mesenchymal transition, producing connective tissue cells that generate ECM components and induce tubular atrophy. Finally, microvascular damage results in the alteration of almost the entire renal parenchyma, amplifying the fibrotic response [93,94].

These increases in renal ECM tend to cause mesangial expansion and proliferation, glomerular hypertrophy, glomerulosclerosis, and tubular basement membrane thickening (TBT), leading to hyperfiltration, altered eGFR, and overall functional and anatomical kidney damage [12,19,20,21,23]. These alterations were observed in our DN groups compared to the control group (I). Studies conducted on STZ-induced diabetic rats, as well as Zucker diabetic fatty (ZDF) and OLETF rats, have shown that the use of pioglitazone for 2, 8, and 10 weeks can promote histopathological recovery, inhibiting tubulointerstitial fibrosis, glomerulosclerosis, as well as hypertrophy and TBT [82,95,96]. Additionally, an animal model of diabetes in STZ-induced Wistar rats proved that 6 weeks of treatment with linagliptin resulted in significant reductions in glomerulosclerosis, tubulointerstitial fibrosis, tubular degeneration, and epithelial atrophy [97]. Furthermore, another study on unilateral right ureteral obstruction (UUO) in C57BL/6 mice treated with tamsulosin for 7 and 14 days showed that renal fibrosis was reduced compared to the UUO group, corroborating the effect of α1-adrenergic receptor antagonism [98].

These increases in kidney ECM tend to cause mesangial expansion and proliferation, glomerular hypertrophy, glomerulosclerosis, and TBT, leading to hyperfiltration, altered eGFR, and overall functional and anatomical kidney damage [12,19,20,21,23]. These changes were detected in our DN groups compared to the control group (I). Studies conducted on STZ-induced diabetic rats, as well as Zucker diabetic fatty (ZDF) and OLETF rats, have shown that the use of pioglitazone for 2, 8, and 10 weeks can promote histopathological recovery, inhibiting tubulointerstitial fibrosis, glomerulosclerosis, as well as hypertrophy and TBT [82,95,96]. Additionally, an animal model of diabetes in STZ-induced Wistar rats showed that 6 weeks of treatment with linagliptin resulted in significant reductions in glomerulosclerosis, tubulointerstitial fibrosis, tubular degeneration, and epithelial atrophy [97]. In addition, another study on UUO in C57BL/6 mice treated with tamsulosin for 7 and 14 days showed that renal fibrosis was decreased compared to the UUO group, supporting the effect of α1-adrenergic receptor antagonism [98].

In this model, the administration of tamsulosin did not achieve recovery of mesangial hypertrophy and expansion. However, it did result in a significant reduction in collagen in the glomerular area, an important marker of glomerulosclerosis, as well as attenuation of TBT and decreases in the presence of collagen I in the tubulointerstitial area. Also, the administration of pioglitazone, both independently and in combination in the LG + PIO and LG + TS + PIO groups, led to the reorganization and restoration of normal glomerular and tubulointerstitial structure, diminishing glomerular hypertrophy, mesangial expansion, as well as the area occupied by collagen I in the glomerular region, and decreasing TBT. Moreover, it significantly reduced interstitial fibrosis with respect to collagen I and III. Concerning linagliptin and the dual combination of LG + TS, they produced effects like tamsulosin, with partial recovery of the parameters. However, a certain degree of hypertrophy and alteration in the oval shape of the glomerulus was still present.

Inflammation plays a crucial role in the progression of DN [12], including the participation of various immune cells such as neutrophils, T lymphocytes, macrophages, among others, as well as the production of chemokines, interleukins, adhesion factors, and more. One of the key factors in this process is the transcription factor NF-κB, which can be activated by several stimuli such as ROS, mechanical stress, cytokine release by immune cells, hyperglycemia, among others [99]. This factor induces the transcription of inflammatory genes that normally act positively by regulating tissue damage. However, when this damage is not resolved in the short term, it contributes to the development and progression of DN [100].

A study conducted on mice overexpressing C-reactive protein (CRP) and treated with linagliptin for 12 weeks showed that the expression of the *p*-65 protein, which is activated in the NF-κB signaling pathway, was significantly decreased in renal tissue [101]. Another study on diabetic rats treated with pioglitazone for 6 months showed, through cortical tissue electrophoresis, that the densitometry of the p65 protein was significantly reduced in the treated groups compared to the type 2 diabetic groups, associating this finding with a notable improvement in histology and functional markers in DN.

One of the target genes of the NF-κB transcription factor is IL-1β, a protein that plays a fundamental role in inflammatory processes in response to infectious agents, some type of injury, or pathological processes, including DN [102]. A study on STZ-induced DN in C57BL/6 mice knockout for apolipoprotein E and treated with pioglitazone for 8 weeks showed significant reductions in serum markers and mesangial expansion due to the inhibition of NF-κB and inflammasome formation, including IL-1β, NLRP3, and caspase 1 [103]. Another study on renal fibrosis induction in C57BL6 mice through double-ligation UUO and treated with gemigliptin, a linagliptin analog, for 14 days [69], showed that tubular atrophy and renal fibrosis were significantly reduced compared to the DN group. Additionally, immunohistochemical and Western blot assays showed decreases in the NF-κB transcription factor, NLRP3, ASC protein, caspase 1, and IL-1β, thus promoting the functional renal recovery of the animals [104].

In response to this inflammatory process, the body attempts to hold the immune response through the activation of anti-inflammatory pathways, with the production of IL-10 being key to this process. IL-10 activates kinases such as Jak1 and Tyk2, leading to the phosphorylation of various transcription factors like STAT1, STAT3, and STAT5, ultimately resulting in the synthesis of anti-inflammatory genes [105].

Studies on DN derived from myocardial infarction in diabetic mice show that treatment with IL-10 led to a reduction in renal collagen-I, attenuating fibrosis [106]. Other studies show that IL-10 expression is a protective factor in DN, inhibiting mesangial proliferation and expansion, as well as reducing inflammatory infiltrate in the renal parenchyma [107]. Additionally, a study conducted on human patients with a diagnosis of type 2 diabetes mellitus (T2DM) treated with sitagliptina, a linagliptin analog [69], for approximately 48 weeks, showed significant increases in IL-10 levels in circulating monocytes in peripheral blood [108]. Furthermore, another murine study involving cecal ligation and puncture (CLP) to induce polymicrobial sepsis demonstrated that administration of pioglitazone 1, 4, and 18 h before CLP resulted in increased IL-10 levels in the peritoneal fluid of treated animals, along with inhibition of inflammation via the Myd88 pathway [109].

Thus, immunofluorescence and qPCR analyses were performed on three factors: NF-κB, IL-1β, and IL-10. Regarding the first inflammatory parameter, immunofluorescence revealed a significant increase in the number of NF-κB-positive cells in the DN group compared to the control group (I), suggesting increased inflammation in the diabetic model. Additionally, the PIO, LG + PIO, and LG + TS + PIO groups showed a significant decrease compared to the DN group, as well as a significant reduction in NF-κB mRNA expression in the PIO and LG + PIO groups. However, the other groups also showed non-significant reductions, except for the LG + TS group, which showed no effect. Furthermore, regarding IL-1β, the administration of tamsulosin, pioglitazone, and linagliptin, as well as their combinations LG + TS, LG + PIO, and LG + TS + PIO, showed significant reductions compared to the diabetic group. Finally, the number of IL-10-positive cells and its gene expression revealed significant decreases in the number of IL-10-positive cells in the DN group due to a lack of anti-inflammatory response caused by DN. This expression was partially recovered by the PIO, LG + PIO, and LG + TS + PIO groups. Additionally, gene expression showed positive regulation by the administration of tamsulosin, pioglitazone, and linagliptin, as well as their combinations LG + TS, LG + PIO, and LG + TS + PIO.

TGF-β is a transcription factor proposed as the main trigger for the development of DN. In both animal and human models, the TGF-β/Smad 2/Smad 3 signaling pathway has been described as playing a crucial role in regulating DN progression by stimulating the production of ECM components such as collagen I, IV, laminin, fibronectin, as well as fibroblast proliferation and epithelial–mesenchymal transition [110,111]. A murine study on animals overexpressing CRP and treated with linagliptin for 12 weeks demonstrated significant reductions in collagen IV compared to the control group (I), associating this with delayed progression of renal damage [101]. Another study on STZ-induced DN in rats showed that 6 weeks of linagliptin administration resulted in significant serum reductions in TGF-β compared to healthy groups [97]. Additionally, a study on T2DM animals treated with pioglitazone for 6 months demonstrated significant reductions in renal collagen IV expression, suggesting morphological and functional recovery of the glomerular basement membrane (GBM). This same study also reported reduced mRNA levels coding for collagen IV and TGF-β, corroborating histological recovery of the renal parenchyma [81]. Finally, another study on UUO in C57BL/6 mice treated with tamsulosin for 7 and 14 days showed significant reductions in α-SMA protein, which plays important roles in cardiac contraction and mechanotransduction. When overexpressed in renal tissue, α-SMA eventually replaces renal parenchyma with scar tissue, with its marker being directly proportional to the number of myofibroblast-producing cells [112]. Thus, tamsulosin reduced fibrosis; this study also showed significant reductions in Smad 3 proteins, directly involved as adaptor proteins in the TGF-β signaling pathway, as determined by Western blot [98].

Therefore, the results of TGF-β mRNA expression, as well as the number of positive cells in immunofluorescence, showed increased levels in the diabetic group compared to the control group (I), suggesting an increase in ECM protein production. However, this condition was negatively regulated by the effect of all the drugs used, as well as their combinations, resulting in significantly lower expression levels compared to the DN group. In this context, collagen IV, a product of TGF-β, also showed significantly higher expression by qPCR compared to the control group (I). Additionally, immunofluorescence revealed thickening and diffusion present in the GBM of the STZ group compared to the control group (I). Consequently, the administration of all the drugs in this study, as well as their combinations, managed to decrease collagen IV mRNA and partially restore GBM integrity.

Oxidative stress plays a vital role in the onset and progression of DN [12], with hyperglycemia being the initiating factor. High glucose levels can induce mitochondrial ROS production via NADPH oxidases, damaging all cell populations in the renal parenchyma. This leads to hyperfiltration, activation of the polyol pathway, generation of advanced glycation end products (AGEs), inhibition of glycolysis, and activation of other pathological signaling pathways (the PKC and hexosamine pathways) [13]. Studies have shown that oxidative stress in DN is associated with mesangial expansion, thickening of the glomerular basement membrane (GBM) and tubular basement membranes, podocyte damage, tubulointerstitial fibrosis, glomerulosclerosis, and cell death [113]. Additionally, recent studies determine that inhibiting oxidative stress in DN animal models has led to comprehensive improvements in clinical signs associated with DN [11,114].

NRF2 is a transcription factor associated with the body’s antioxidant defense. Through its interaction with the ROS sensor protein Keap1 (Kelch-like ECH-associated protein 1), NRF2 activation and later translocation to antioxidant response element (ARE) promoter regions occur. This induces the transcription of genes related to detoxification and antioxidant protection, such as the enzymes NQO1 (quinone oxidoreductase 1), GCL (γ-glutamyl-cysteine), GSH (glutathione), and HO-1. HO-1 catalyzes the conversion of heme into ferrous ion, biliverdin, and bilirubin, the latter two possessing antioxidant and anti-inflammatory properties [115].

A study on the induction of acute pancreatitis with lipopolysaccharide and caerulein, aimed at developing acute lung injury in a mouse model, observed that treatment with sitagliptin, a linagliptin analog, reduced ROS via stimulation of the p62–Keap1–Nrf2 signaling pathway and its synthesis products such as HO-1 [116]. Another study conducted on a hypertensive rat model treated with pioglitazone for 10 days demonstrated a positive stimulation of NRF2, associating it with improvements in the regulation of normal blood pressure levels [76]. Additionally, another study on tamoxifen-induced hepatotoxicity in rats showed that pioglitazone administration for 10 days regulated the Keap1/Nrf2/HO-1 signaling pathway, providing hepatoprotection by reducing apoptosis, as well as inflammatory and oxidative activity [117].

Thus, the results of HO-1 and NRF2 mRNA gene expression in this DN model showed significant increases with the administration of tamsulosin, pioglitazone, and linagliptin, as well as the combinations LG + TS, LG + PIO, and LG + TS + PIO, compared to the DN group (*p* < 0.05, *p* < 0.01, *p* < 0.001, *p* < 0.0001). The DN group showed a notable decline in antioxidant defense compared to the control group (I), indicating a potential depletion in this mechanism. However, this depletion was reversed by the administration of tamsulosin, pioglitazone, and linagliptin, as well as their combinations, resulting in an outcome statistically comparable to the control group.

In summary, the administration of tamsulosin, pioglitazone, and linagliptin, as well as their combinations (LG + TS, LG + PIO, and LG + TS + PIO) for a period of six weeks, demonstrated renoprotective effects in the development of DN. Pioglitazone exhibited a more pronounced inhibitory effect on the progression of DN compared to tamsulosin and linagliptin, which demonstrated comparable activity. Therefore, these drugs could potentially have significant potential in generating new effective therapeutic strategies for improving DN.

## 4. Materials and Methods

### 4.1. Animals

A total of 40 male Wistar rats (6 ± 1 weeks, 180–200 g) were obtained from the animal facility of the Universidad Autónoma de Aguascalientes (UAA, Mexico). The rats were housed and maintained under sterile, pathogen-free conditions with 12 h light/dark cycles, a relative humidity of 50–60%, and a constant temperature of 20–25 °C. They were fed a standard rodent diet (*ad libitum*), supplemented with 50 g of food (Purina^®^ DOG CHOW mature age, Nestlé, Mexico City, Mexico). The animals were hydrated with purified water containing 1% sucrose, with both food and water provided ad libitum. All animal experiments were approved by the Ethics Committee for the use of animals in teaching and research at UAA (CEADI-UAA, UAA: Universidad Autónoma de Aguascalientes, AUT-B-C-1121-077-Type C) following the Mexican Official Standard NOM-062-ZOO-1999 [118], and the guidelines of the National Institutes of Health for the care and use of laboratory animals [119]. During the experiment, various signs related to humane endpoints were checked weekly, including stress, pain, decreased mobility, withdrawal, weight loss, reduced water and food intake, self-mutilation, and behavioral changes such as aggression.

### 4.2. Experimental Design

After a one-week acclimatization period, the DM induction protocol described by Furman et al. [63], with modifications, was followed. The rats were fasted for one day and then administered a single high dose of STZ (Sigma Aldrich Biotechnology, St. Louis, MO, USA) intraperitoneally (50 mg/kg), dissolved in a freshly prepared 0.1 mol/L citrate buffer solution at pH 4.5. Rats with fasting glucose levels greater than 13.9 mmol/L (250 mg/dL) [65], three days after the STZ injection were diagnosed as diabetic. The rats were randomly divided into 8 groups (*n* = 5): (I) the intact group (I); (II) the DN group or positive control, induced with STZ (DN); (III) the DN group treated with tamsulosin (diluted in purified water at a dose of 0.4 mg/kg, administered intragastrically, Pharmalife, Mexico, Mexico) (TS); (IV) the DN group treated with pioglitazone (diluted in 0.5% carboxymethylcellulose (CMC) at a dose of 60 mg/kg, administered intragastrically, Pharmalife, Jalisco, Mexico) (PG); (V) the DN group treated with linagliptin (diluted in purified water at a dose of 5 mg/kg, administered intragastrically, Boehringer Ingelheim, Mexico City, Mexico) (LG); (VI) the DN group treated with linagliptin and tamsulosin simultaneously (LG + TS); (VII) the DN group treated with linagliptin and pioglitazone simultaneously (LG + PG); (VIII) the DN group treated with linagliptin, tamsulosin, and pioglitazone simultaneously (LG + TS + PG). Weekly, the animals’ weight, water and food consumption, and fasting blood glucose (FBG) were measured using Accu-Chek, Performa (Figure 11).

After the treatment period, the rats were euthanized (≥100 mg/kg sodium pentobarbital) via intraperitoneal injection. A thoracotomy was then performed, and blood samples were collected by cardiac puncture. The blood was centrifuged at 1500× *g* for 15 min to obtain serum, which was stored at −80 °C for subsequent biochemical marker analysis. Urine samples were also collected and stored at −20 °C. Finally, kidney tissue samples were obtained for further studies.

### 4.3. Biochemical Parameters

The concentrations of glucose, BUN, urea, and creatinine in serum and urine were measured using wet chemistry, employing the Siemens Dimension EXL-200 equipment, following the manufacturer’s instructions.

### 4.4. Renal Index (KI)

The freshly extracted kidneys were weighed separately to calculate the renal index, using the following formula mentioned by Qi et al [11].
KI = KW/BW,(1)
where “KI” is the renal index measured in mg/g, “KW” is the kidney weight measured in mg, and “BW” is the body weight of the rat measured in g.

### 4.5. Estimated Glomerular Filtration Rate (eGFR)

The eGFR was estimated using the formula mentioned by Besseling et al. [120]
Plasma creatinine < 52 μmol/L: eGFR = 880 × W^0.695^ × C^−0.660^ × U^−0.391^,Plasma creatinine ≥ 52 μmol/L: eGFR = 5862 × W^0.695^ × C^−1.150^ × U^−0.391^(2)
where “eGFR” is the estimated glomerular filtration rate measured in ml/min, “W” is the body weight of the rat measured in g, “C” is the creatinine concentration measured in μmol/L, and “U” is the urea concentration measured in mmol/L.

### 4.6. Histopathology

After 2 days of fixation in 10% neutral formalin solution, the kidneys were dehydrated in increasing concentrations of alcohol for 1 h each (96%, 96%, 100%, 100%), followed by clearing with xylene (2 changes for 1 h each) and infiltration with paraffin (2 changes for 2 h each). Histological sections of 5 μm were prepared and stained with H&E, Masson’s trichrome, Sirius red, and PAS. Image capture was performed using an inverted optical microscope (ZEISS-Axiovert 40 CFL) for all the stains, adding polarizing lenses for the analysis of tissues stained with Sirius red. A total of 2 fields (400× for H&E, Masson’s trichrome, PAS, and 200× for Sirius red) per tissue per slide were analyzed. The analysis was conducted using Fiji-Image Pro Plus software to perform morphometric measurements of the ratio of the urinary space area to the total renal corpuscle area, expressed as a percentage, for the analysis of glomerular hypertrophy and mesangial expansion in H&E staining using the following formula
% of USP = USA/RCA × 100 (3)
where “USP” is the percentage of the urinary space, “USA” is the area of the urinary space in μm^2^, and “RCA” is the area of the renal corpuscle in μm^2^.

Additionally, the area of tubulointerstitial fibrosis relative to collagen I (red) and III (green) in Sirius red staining was analyzed, as mentioned by Song et al. [65]. The fraction of the total glomerular collagen volume (glomerulosclerosis) in Masson’s trichrome staining was also analyzed, as performed by Fujisawa et al. [121], as well as the thickening of the glomerular basement membrane through PAS staining positivity.

### 4.7. Immunofluorescence

Pathological samples fixed in paraformaldehyde were sectioned into 1 cm^3^ pieces and placed in 15% and 30% sucrose solutions overnight for two days, respectively. Then, 5 μm thick sections were made using a cryostat (Walldorf, Germany, ZEISS, HYRAX C 2′) and fixed on a hot plate for 5 min. The sections were then washed for 10 min in a 1× PBS solution. Next, nonspecific epitope binding was blocked with 10% fetal bovine serum in 1× PBS for 30 min at 37 °C, and the slides were placed in a humid chamber.

For staining, primary antibodies (Massachusetts, USA, TGF-β: Invitrogen, No. TB21; Missouri, USA, Collagen IV: NOVUS, No. NB120-6586; Massachusetts, USA, pNF-κB p65: Invitrogen, No. Ser529 #44-711G; Cambridge, United Kingdom, IL-10: Abcam, No. AB34843; Cambridge, United Kingdom, IL-1β: NOVUS, No. NB600-633) were used at a 1:100 dilution in 1× PBS and incubated overnight in a humid chamber at 4 °C. After three 5 min washes with 1× PBS solution, a secondary anti-rabbit IgG antibody conjugated with Alexa Fluor (California, USA, Life Technologies, Ref: A11008) was used at a 1:1000 dilution in 1× PBS, incubated for 1 h at 37 °C and 30 min at room temperature, in the dark and in a humid chamber. Finally, counterstaining was performed with Hoechst solution for nuclei (Sigma-Aldrich) for 10 min, followed by mounting with Glicergel (Glostrup, Denmark, C0563-Dako).

A total of 2 fields (X400 for Collagen IV, and X200 for NF-κB, IL-10, IL-1β, and TGF-β) per slide were analyzed using a fluorescence microscope (ZEISS- Axio Vision Release 4.6.3 SP1) with UV light filters. Image processing was performed using Fiji-Image Pro Plus software to determine the positive area for Collagen IV, as well as the number of activated cells for IL-10 and pNF-κB p65.

### 4.8. Quantitative Real-Time PCR

mRNA was extracted from renal samples using the Direct-zol™ RNA MiniPrep kit (Zymo Research, R2050) according to the manufacturer’s specifications. The quantity and purity of the mRNA were then determined using a Biodrop device (Isogen Life Science, 80-3006-51). Next, reverse transcription was performed with 1 μg of mRNA using the GoScript™ Reverse Transcription System (Fitchburg, WI, USA, Promega Corporation, A5000) and a thermocycler (Swift™ MiniPro^®^). Finally, qPCR was conducted using Maxima SYBR Green/ROX qPCR Master Mix (2X) (Plainville, MA, USA, Thermo Scientific, K0221) on an Applied Biosystems StepOne Real-Time PCR System, utilizing the corresponding primers (Table 1) for TGF-β, collagen IV, pNF-κB p65, IL-10, IL-1β, HO-1, and NRF2 (IBT© 2020 Integrated DNA Technologies, Inc.). Relative expression levels were quantified in triplicate and normalized against the β-Actin gene as a housekeeping gene, with differences determined using the relative ΔΔCt method.

### 4.9. Statistical Analyses

Data were analyzed using GraphPad Prism 9 software (San Diego, CA, USA). One-way analysis of variance (ANOVA) was used to evaluate data from multiple groups for unpaired samples, followed by Tukey’s post hoc test. Data were considered statistically significant when *p* < 0.05.

## 5. Conclusions

The administration of tamsulosin, pioglitazone, and linagliptin, as well as their combinations, over a period of 6 weeks in a STZ-induced DN model in Wistar rats, demonstrated a restorative effect on the morphological changes associated with DN. In addition, a partial recovery of renal function has been observed, along with a reduction in fibrosis. This was evidenced by the inhibition of TGF-β and collagen IV, which exerted anti-inflammatory effects by inhibiting NF-κB and IL-1β and stimulating IL-10. Furthermore, antioxidant effects were demonstrated through the activation of NRF2.

## Figures and Tables

**Figure 1 ijms-25-11372-f001:**
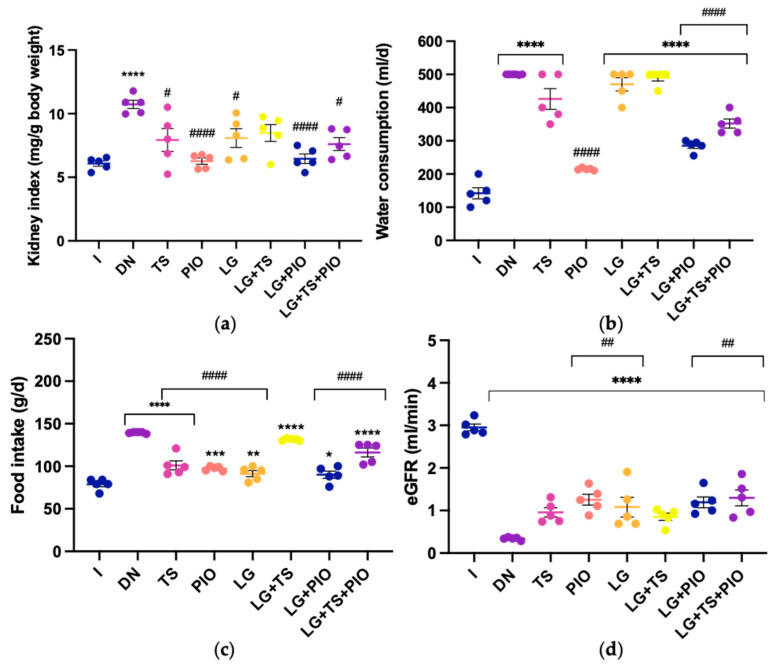
The PIO and LG + PIO groups saw improved KI, water and food intake, and eGFR in rats with DN. (**a**) PIO and LG + PIO restored renal index levels; (**b**) PIO, LG + PIO, and LG + TS + PIO reduced polydipsia in diabetic rats; (**c**) all treatments except LG + TS reduced polyphagia in DN rats; (**d**) all treatments increased eGFR levels. I; DN; TS; PIO; LG; LG + TS; LG + PIO; LG + TS + PIO. Data are presented as mean ± standard error of mean; *n* = 5. * *p* < 0.05 vs. I group, ** *p* < 0.01 vs. I group, *** *p* < 0.001 vs. I group, **** *p* < 0.0001 vs. I group, # *p* < 0.05 vs. DN group, ## *p* < 0.01 vs. DN group, #### *p* < 0.0001 vs. DN group.

**Figure 2 ijms-25-11372-f002:**
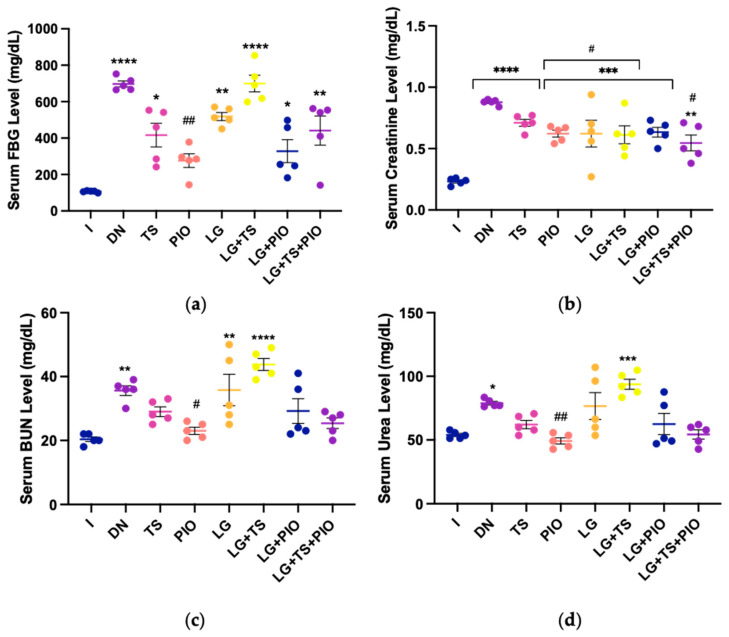
The PIO group saw improved serum renal function markers in rats with DN. (**a**) PIO significantly reduced FBG in rats with DN; (**b**) PIO, LG, LG + TS, and LG + TS + PIO showed reduced serum creatinine levels; (**c**) the PIO group partially restored normal serum BUN levels; (**d**) the PIO group restored normal serum urea levels. I; DN; TS; PIO; LG; LG + TS; LG + PIO; LG + TS + PIO. Data are presented as mean ± standard error of mean; *n* = 5. * *p* < 0.05 vs. I group, ** *p* < 0.01 vs. I group, *** *p* < 0.001 vs. I group, **** *p* < 0.0001 vs. I group, # *p* < 0.05 vs. DN group, ## *p* < 0.01 vs. DN group.

**Figure 3 ijms-25-11372-f003:**
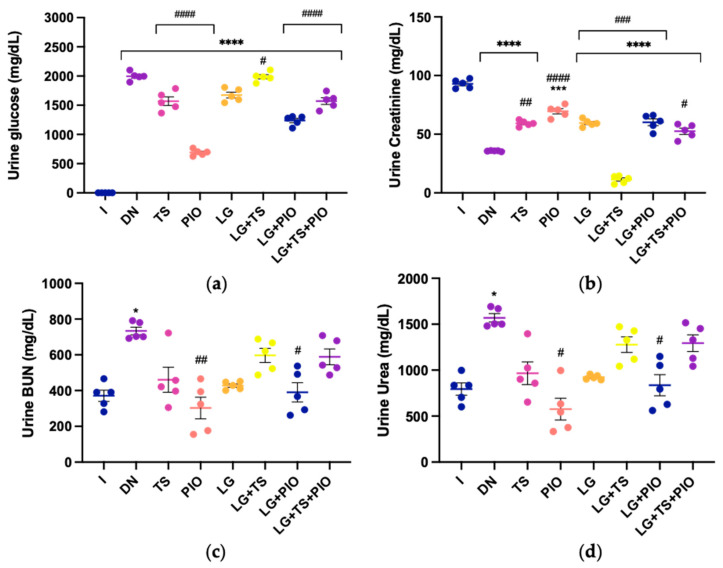
The PIO and LG + PIO groups saw improved urinary markers of renal damage in rats with DN. (**a**) The PIO, TS, and LG + PIO groups reduced glucosuria; (**b**) TS, PIO, LG, LG + PIO, and LG + TS + PIO significantly increased creatinine levels; (**c**) PIO and LG + PIO decreased urinary BUN excretion; (**d**) PIO and LG + PIO decreased urinary urea excretion. I; DN; TS; PIO; LG; LG + TS; LG + PIO; LG + TS + PIO. Values are presented as mean ± standard error of mean; *n* = 5. * *p* < 0.05 vs. I group, *** *p* < 0.001 vs. I group, **** *p* < 0.0001 vs. I group, # *p* < 0.05 vs. DN group, ## *p* < 0.01 vs. DN group, ### *p* < 0.001 vs. DN group, #### *p* < 0.0001 vs. DN group.

**Figure 4 ijms-25-11372-f004:**
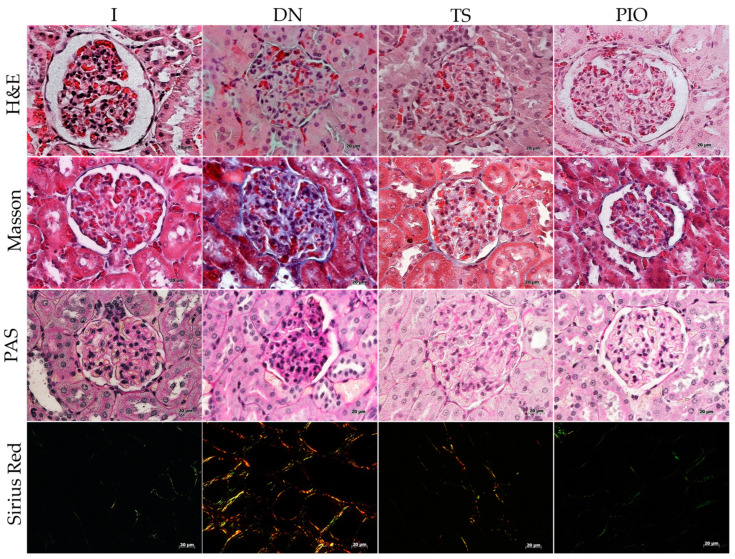
The histopathological analysis of the effect of tamsulosin, pioglitazone, and linagliptin in rats with DN. (**a**) The PIO, LG, LG + PIO, and LG + TS + PIO groups saw increases in the urinary space. The PIO, LG + PIO, and LG + TS + PIO groups saw reduced collagen I deposition. The PIO and LG + PIO groups saw restored GBM structure. The PIO, LG + PIO, and LG + TS + PIO groups saw decreased collagen I and III deposition in the tubulointerstitial space. Morphometric analysis: (**b**) urinary space, (**c**) collagen volume fraction, (**d**) PAS-positive area, (**e**) collagen type I-positive area, and (**f**) collagen type III-positive area (Sirius red). I, DN, TS, PIO, LG, LG + TS, LG + PIO, LG + TS + PIO. H&E staining. Masson’s trichrome. PAS and Sirius red. Magnification (×40) (Scale bar = 20 μm). Data are presented as mean ± standard error of mean; *n* = 5, 2 fields per slide. ** *p* < 0.01 vs. group I, *** *p* < 0.001 vs. group I, **** *p* < 0.0001 vs. group I, # *p* < 0.05 vs. DN group, ## *p* < 0.01 vs. DN group, ### *p* < 0.001 vs. DN group, #### *p* < 0.0001 vs. DN group.

**Figure 5 ijms-25-11372-f005:**
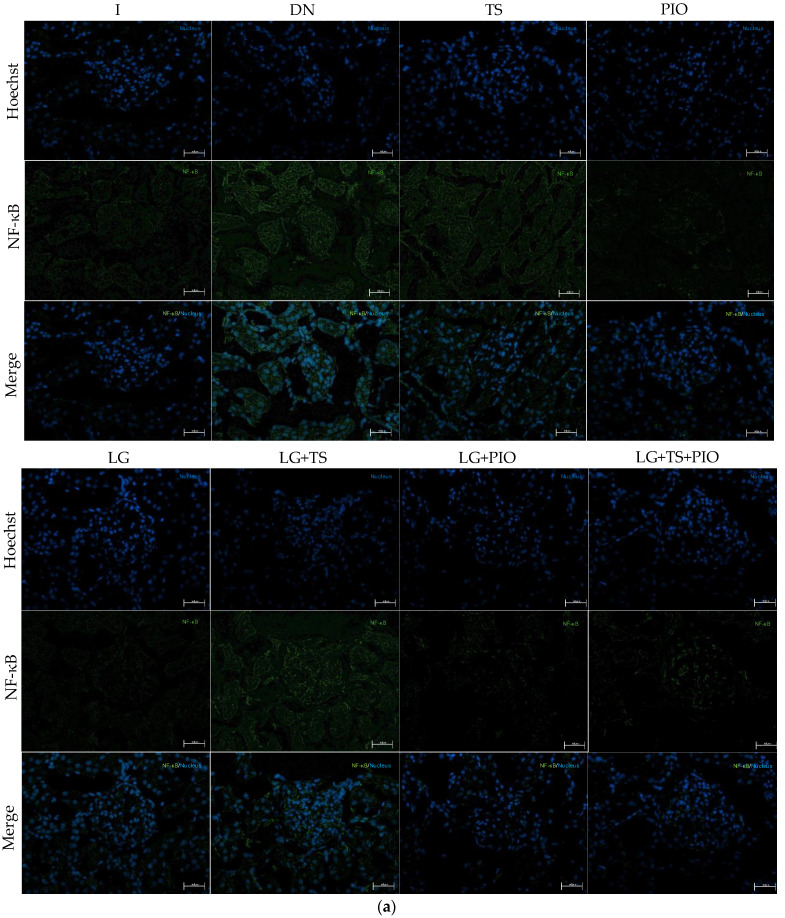
The immunofluorescence and gene expression of NF-κB. (**a**) The PIO, LG + PIO, and LG + TS + PIO groups saw reduced NF-κB positivity in DN rats (×200) (calibration bar = 50 μm); (**b**) morphometric analysis of the number of positive cells per square millimeter, 2 fields per slide were analyzed; (**c**) the relative gene expression of NF-κB mRNA. I; DN; TS; PIO; LG; LG + TS; LG + PIO; LG + TS + PIO. Data are presented as mean ± standard error of mean; *n* = 5. * *p* < 0.05 vs. I group, *** *p* < 0.001 vs. I group, # *p* < 0.05 vs. DN group, ## *p* < 0.01 vs. DN group, ### *p* < 0.001 vs. DN group.

**Figure 6 ijms-25-11372-f006:**
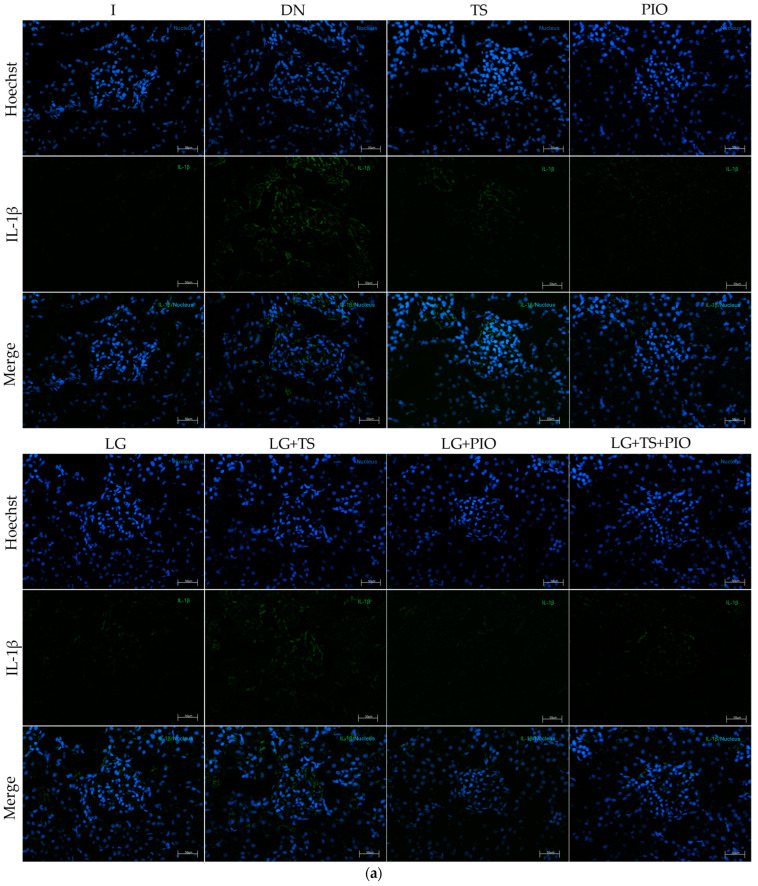
The immunofluorescence and gene expression of IL-1β. (**a**) The PIO and LG + PIO groups saw reduced IL-1β positivity in DN rats (X200) (calibration bar = 50 μm); (**b**) morphometric analysis of the number of positive cells per square millimeter, 2 fields per slide were analyzed; (**c**) the relative gene expression of IL-1β mRNA. I; DN; TS; PIO; LG; LG + TS; LG + PIO; LG + TS + PIO. Data are presented as mean ± standard error of mean; *n* = 5. * *p* < 0.05 vs. I group, ** *p* < 0.01 vs. I group, **** *p* < 0.0001 vs. I group, # *p* < 0.05 vs. DN group, ### *p* < 0.001 vs. DN group, #### *p* < 0.0001 vs. DN group.

**Figure 7 ijms-25-11372-f007:**
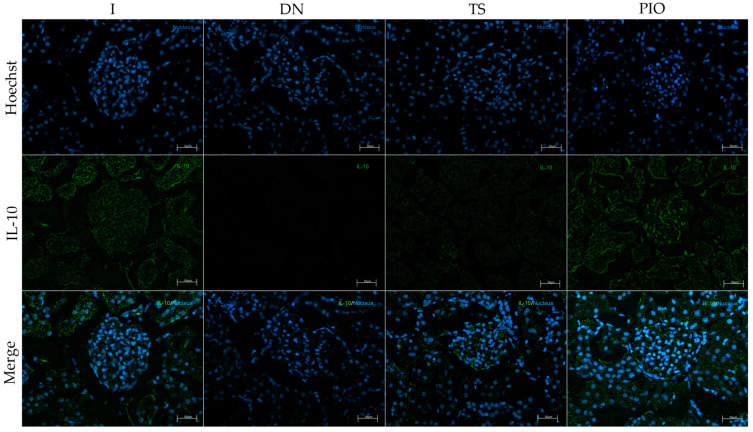
The immunofluorescence and gene expression of IL-10. (**a**) The PIO, LG + PIO, and LG + TS + PIO groups saw increased IL-10 positivity in DN rats (X200) (calibration bar = 50 μm); (**b**) morphometric analysis of the number of positive cells per square millimeter, 2 fields per slide were analyzed; (**c**) the relative gene expression of IL-10 mRNA. I; DN; TS; PIO; LG; LG + TS; LG + PIO; LG + TS + PIO. Data are presented as mean ± standard error of mean; *n* = 5. * *p* < 0.05 vs. I group, ** *p* < 0.01 vs. I group, *** *p* < 0.001 vs. I group, **** *p* < 0.0001 vs. I group, ## *p* < 0.01 vs. DN group, ### *p* < 0.001 vs. DN group, #### *p* < 0.0001 vs. DN group.

**Figure 8 ijms-25-11372-f008:**
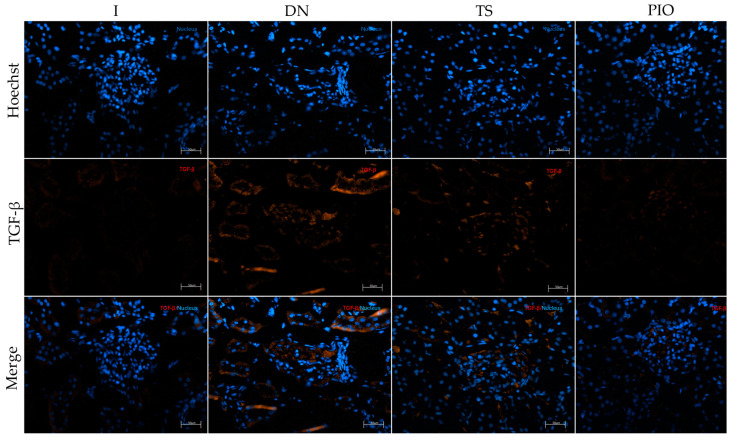
The immunofluorescence and gene expression of TGF-β. (**a**) The PIO and LG + PIO groups reduced TGF-β positivity in DN rats (×200) (calibration bar = 50 μm); (**b**) morphometric analysis of the number of positive cells per square millimeter, 2 fields per slide were analyzed; (**c**) the relative gene expression of TGF-β mRNA. I; DN; TS; PIO; LG; LG + TS; LG + PIO; LG + TS + PIO. Data are presented as mean ± standard error of mean; *n* = 5. ** *p* < 0.01 vs. I group, **** *p* < 0.0001 vs. I group, # *p* < 0.05 vs. DN group, ## *p* < 0.01 vs. DN group, #### *p* < 0.0001 vs. DN group.

**Figure 9 ijms-25-11372-f009:**
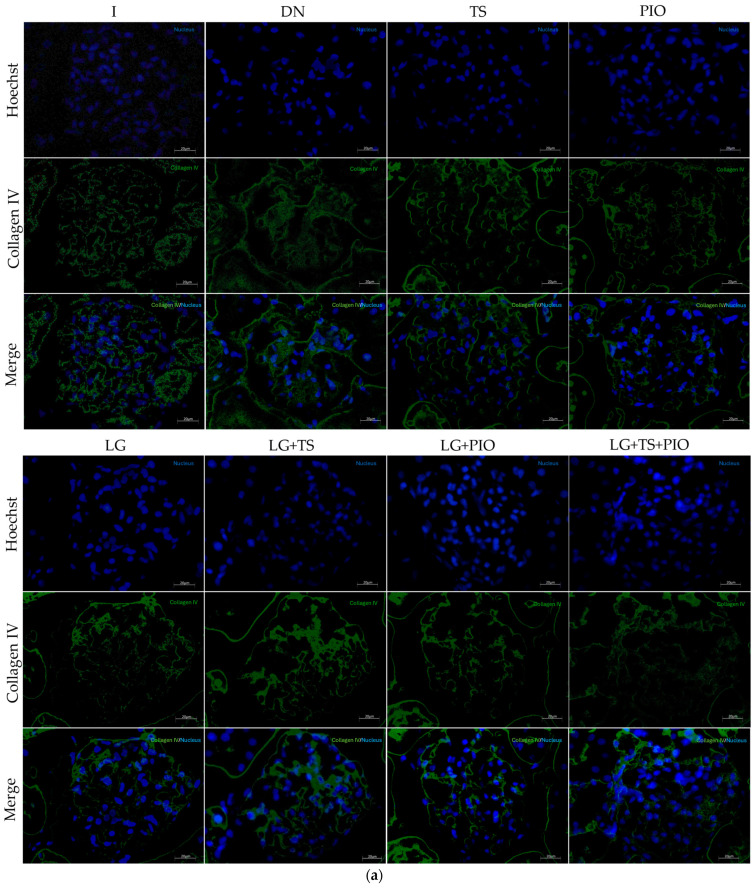
The immunofluorescence and gene expression of Col-IV. (**a**) The PIO, LG + PIO, and LG + TS + PIO groups saw a reduced percentage of Col-IV positivity in DN rats (X200) (calibration bar = 50 μm); (**b**) morphometric analysis of the Col-IV positive area, 2 fields per slide were analyzed; (**c**) relative gene expression of Col-IV mRNA. I; DN; TS; PIO; LG; LG + TS; LG + PIO; LG + TS + PIO. Data are presented as mean ± standard error of mean; *n* = 5. * *p* < 0.05 vs. I group, ** *p* < 0.01 vs. I group, **** *p* < 0.0001 vs. I group, # *p* < 0.05 vs. DN group, ## *p* < 0.01 vs. DN group, ### *p* < 0.001 vs. DN group, #### *p* < 0.0001 vs. DN group.

**Figure 10 ijms-25-11372-f010:**
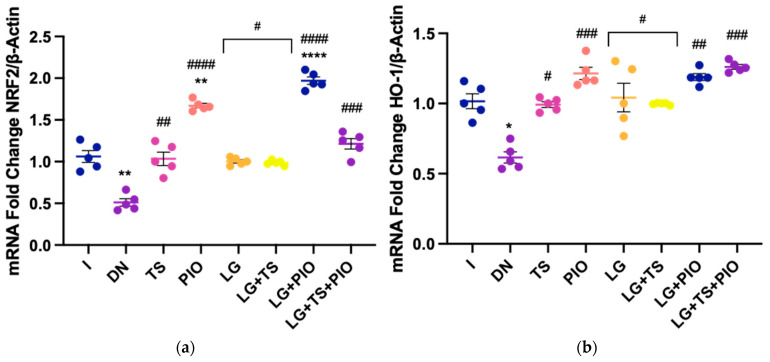
The gene expression of NRF2 and HO-1. (**a**) The PIO and LG + PIO groups saw stimulated NRF2 gene expression in DN rats; (**b**) the PIO, LG + PIO, and LG + TS + PIO groups saw stimulated HO-1 gene expression in DN rats. I; DN; TS; PIO; LG; LG + TS; LG + PIO; LG + TS + PIO. Data are presented as mean ± standard error of mean; *n* = 5. * *p* < 0.05 vs. I group, ** *p* < 0.01 vs. I group, **** *p* < 0.0001 vs. I group, # *p* < 0.05 vs. DN group, ## *p* < 0.01 vs. DN group, ### *p* < 0.001 vs. DN group, #### *p* < 0.0001 vs. DN group.

**Figure 11 ijms-25-11372-f011:**
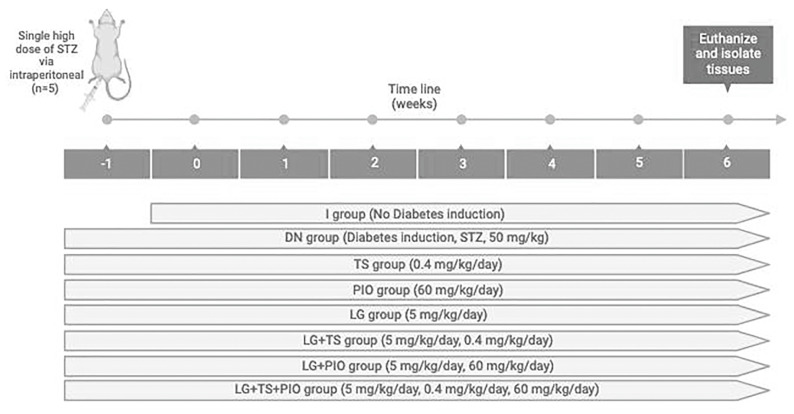
Experimental Design. Displaying the application of STZ and treatments with tamsulosin, pioglitazone, and linagliptin.

**Table 1 ijms-25-11372-t001:** Primer Sequence for qPCR.

Gen	Sequence	pb
TGF-β	Fw: 5′-GAC TCT CCA CCT GCA AGA CCA-3′	21
	Rv: 5′-CGG GTG ACT TCT TTG GCG TA-3′	20
Col-IV	Fw: 5′-TGC CTT ACA GGG ATT TGC GT-3′ Rv: 5′-GTG TGC CAT TAT GGG AGG CT-3′	20 20
NF-κB	Rv: 5′-CGG GTG ACT TCT TTG GCG TA-3′	20
	Rv: 5′-CAC ACA GAA TGA GGC TTA TTC C-3′	22
IL-10	Fw: 5′-TGG CTC AGC ACT GCT AGT TT-3′	20
	Rv: 5′-TTG TCC AGC TGG TCC TTC TT-3′	20
HO-1	Fw: 5′-GAA GAG GAG ATA GAG CGA AAC A-3′	22
	Rv: 5′-CAA TCT TCT TCA GGA CCT GAC C-3′	22
IL-1β	Fw: 5’-CTG TGA CTC GTG GGA TGA TG-3’	20
	Rv:5’-GGG ATT TTG TCG TTG CTT GT-3	20
NRF2	Fw: 5′-CAG TCT TCA CCA CCC CTG AT-3′	20
	Rv: 5′-CAG TGA GGG GAT CGA TGA GT-3′	20
β-Actin	Fw: 5′-GTC GTA CCA CTG GCA TTG TG-3′	20
	Rv: 5′-GCT GTG GTG GTG AAG CTG TA-3′	20

TGF-β: transcription grow factor β; Col-IV: collagen IV, NF-κB: nuclear factor kappa light. Chain enhancer of activated B-cells; IL-10: Interleukin 10; IL-1β: Interleukin 1 β, HO: Hemoxigenase 1; NRF2: nuclear factor erythroid 2-related factor 2.

## Data Availability

The data presented in this study are available on request from the corresponding author due to privacy.

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
