# Peer review of "Evaluation of the Effect of an α-Adrenergic Blocker, a PPAR-γ Receptor Agonist, and a Glycemic Regulator on Chronic Kidney Disease in Diabetic Rats"

_ijms, 2024, doi:10.3390/ijms252111372_

Round 1

Reviewer 1 Report

Comments and Suggestions for Authors

Research article entitled ‘Evaluation of the Effect of an α-Adrenergic Blocker, a PPAR-γ Receptor Agonist, and a Glycemic Regulator on Chronic Kidney Disease in Diabetic Rats’ was well received.

The authors have studied the solo and combined effects of pioglitazone, linagliptin, and tamsulosin on kidney functions in preclinical mice models. As the protective role of pioglitazone and linagliptin in kidney functions is already known, they were used as controls for the individual and combined effects of tamsulosin. Here are certain things based on which I don’t recommend publication of this study.

1.       First of all, the study design is flawed. According to the literature, tamsulosin is used to treat benign prostate hyperplasia. The authors have stated in the introduction section and I quote, ‘Oxidative stress plays a crucial role in the progression of DN because hyperglycemia induces the formation of advanced glycation end products (AGEs) [25], leading to the subsequent release of mediators such as angiotensin II and TGF-β, which have been reported to have a significant impact on DN [26,27]. Additionally, hyperglycemia induces the production of reactive oxygen species (ROS) [28].’

And it is stated in the literature that ‘tamsulosin itself induced hyperglycemia’ https://doi.org/10.2337%2Fcd21-0018 . Tamsulosin is a selective alpha-1 adrenergic receptor antagonist. By blocking alpha-1 receptors, tamsulosin might indirectly affect the balance of insulin and glucagon, potentially leading to altered glucose metabolism in susceptible individuals. Although tamsulosin is not a direct cause of hyperglycemia, it could potentially exacerbate existing issues with glucose regulation. So, the use of this drug in combination with anti-diabetics will cause hyperglycemia and oxidative stress, which, according to authors, may significantly affect DN.

2.       Fig.3A, the tamsulosin is demonstrated to decrease urine glucose levels, which contradicts its own functions. 

i think the study design is seriously flawed. 

Author Response

Thank you for reading our article. We appreciate your suggestions on how to improve this work.

Comments. 1:

Research article entitled ‘Evaluation of the Effect of an α-Adrenergic Blocker, a PPAR-γ Receptor Agonist, and a Glycemic Regulator on Chronic Kidney Disease in Diabetic Rats’ was well received.

The authors have studied the solo and combined effects of pioglitazone, linagliptin, and tamsulosin on kidney functions in preclinical mice models. As the protective role of pioglitazone and linagliptin in kidney functions is already known, they were used as controls for the individual and combined effects of tamsulosin. Here are certain things based on which I don’t recommend publication of this study.

  1. First of all, the study design is flawed. According to the literature, tamsulosin is used to treat benign prostate hyperplasia. The authors have stated in the introduction section and I quote, ‘Oxidative stress plays a crucial role in the progression of DN because hyperglycemia induces the formation of advanced glycation end products (AGEs) [25], leading to the subsequent release of mediators such as angiotensin II and TGF-β, which have been reported to have a significant impact on DN [26,27]. Additionally, hyperglycemia induces the production of reactive oxygen species (ROS) [28].’

And it is stated in the literature that ‘tamsulosin itself induced hyperglycemia’ https://doi.org/10.2337%2Fcd21-0018. Tamsulosin is a selective alpha-1 adrenergic receptor antagonist. By blocking alpha-1 receptors, tamsulosin might indirectly affect the balance of insulin and glucagon, potentially leading to altered glucose metabolism in susceptible individuals. Although tamsulosin is not a direct cause of hyperglycemia, it could potentially exacerbate existing issues with glucose regulation. So, the use of this drug in combination with anti-diabetics will cause hyperglycemia and oxidative stress, which, according to authors, may significantly affect DN.

Response   1:

We have reviewed the literature and indeed, a potential hyperglycemic effect of tamsulosin has been observed in a clinical case where the drug was administered after the disease had already been established. However, other studies (https://doi.org/10.1080/21655979.2021.1955527) have reported that tamsulosin, in the presence of high glucose concentrations, can mitigate glucose-induced damage by reducing the expression of TNF-α, IL-6, IL-8, MMP-2, and MMP-9, as well as by decreasing ROS generation and preventing p38 activation.

The aim of our study does not specifically focus on the use of tamsulosin alone; rather, we analyzed the effects of three drugs administered individually or in combination to evaluate their anti-inflammatory, anti-fibrotic, renal function improvement, and potential hypoglycemic effects. The drugs were administered simultaneously with streptozotocin.

In this context, we agree with your comment. However, regarding tamsulosin, anti-fibrotic effects have been reported in various experimental models (https://doi.org/10.1096/fj.202000737RRR, https://doi.org/10.1681/ASN.2012070678), as well as anti-inflammatory effects (https://doi.org/10.1080/21655979.2021.1955527).

In this study, we report data from an experimental model examining the effects of tamsulosin, pioglitazone (https://doi.org/10.1016/j.redox.2021.102029) and linagliptin (https://doi.org/10.12122/j.issn.1673-4254.2023.12.09). Our findings indicate that there is no significant increase in serum glucose levels compared to the streptozotocin-induced group (Figure 2a).

Comments   2: Fig.3A, the tamsulosin is demonstrated to decrease urine glucose levels, which contradicts its own functions

Response    2:

It is important to note that urine glucose levels are primarily a result of glomerular injury. Therefore, the observed decrease in urine glucose suggests an improvement in glomerular function with the use of tamsulosin (https://doi.org/10.1080/21655979.2021.1955527).

Corresponding authors:

Dr. Javier Ventura Juárez

Dra. Sandra Luz Martínez Hernández.

Reviewer 2 Report

Comments and Suggestions for Authors

Morones et al showed that the administration of tamsulosin, pioglitazone, and linagliptin, as well as their combinations, over a period of 6 weeks in a STZ-induced diabetic nephropathy demonstrated restorative effect on the morphological changes associated with DN. However some improvements should be done.

Figure 4 too big, too much information and very small space to see all the data presented.

Figure 5-9. IF images are too small

Scatter dot plots should be used to indicate to the reader the n used and the dispersion inside the groups.

Color for different groups for the graphs are encouraged.

Comments on the Quality of English Language

A minor revision is necessary

Author Response

Thank you for reading our article. We appreciate your suggestions on how to improve this work.

Comments 1:Figure 4 too big, too much information and very small space to see all the data presented.

Response   1:

In Figure 4, we have taken your advice into account. We have enlarged the figures and divided them into sections.

Comments 2: Figure 5-9. IF images are too small.

Response  2: We have enlarged the figures and divided them into sections in Figures 5-9, following  your instructions.

Comments   3: Scatter dot plots should be used to indicate to the reader the n used and the dispersion inside the groups.

Response. 3: We have changed the format of all the graphs and also included the n used.

Comments 4: Color for different groups for the graphs are encouraged.

Response 4: Your instructions have been followed.

Comments 5: Comments on the Quality of English Language. A minor revision is necessary.

Response.  5: A quality check in English has been carried out on the entire article.

Corresponding authors:

Dr. Javier Ventura Juárez

Dra. Sandra Luz Martínez Hernández.

Reviewer 3 Report

Comments and Suggestions for Authors

Diabetes still represents a major health problem and due to its increased prevalence remains a pathology under debate in order to develop new forms of treatment and/or prevention methods. Recently, tamsulosin mechanism of action gained some interest regarding its beneficial impact on the development and progression of diabetic kidney disease. Considering its globally use in prostatic hyperplasia, a pathology also often noticed in daily practice as DM, if tamsulosin role in DKD progression will be confirmed, it will be easy to be recommended and without additional costs. The methodology and findings were well described, and the conclusions were in accordance with the assessed results. One minor suggestion: please better explain in the introduction the correlation between tamsulosin and DN onset.

Author Response

Thank you for reading our article. We appreciate your suggestions on how to improve this work.

Comments 1:  One minor suggestion: please better explain in the introduction the correlation between tamsulosin and DN onset.

Response   1:  We have added a better explanation of how tamsulosin correlates to DN in the Introduction, lines 91-102.

Corresponding authors:

Dr. Javier Ventura Juárez

Dra. Sandra Luz Martínez Hernández.

Round 2

Reviewer 1 Report

Comments and Suggestions for Authors

The revised version was well received. The authors have diligently improved the manuscript. And have addressed the raised points.

I appreciate the authors for providing a detailed response.